# The Arctic Ocean is a net sink for anthropogenic lead deposited into the Atlantic Ocean

Stephan Krisch [1,2] ✉, Arianna Olivelli[3,4,9], Loes J. A. Gerringa[5], Rob Middag [5,6], Birgit Rogalla [7,8] & Eric P. Achterberg [2]

Humans emitted millions of tons of the toxic element lead (Pb) into the atmosphere. The North Atlantic Ocean has been strongly affected by atmospheric Pb deposition, however the role of ocean currents in dispersing the Atlantic dissolved Pb (dPb) burden remains unclear. Here, we show that the Arctic Ocean received a dPb flux of 611 ± 74 Mg·a$^{-1}$ from the North Atlantic Ocean in 2015/2016, making the Arctic Ocean a previously unrecognized net sink of Atlantic dPb (378 ± 85 Mg·a$^{-1}$). This input is comparable to Arctic riverine dPb discharge (344 ± 222 Mg·a$^{-1}$). Lead isotope measurements trace the origin of dPb in the Arctic Ocean back to anthropogenic emissions from North America and Eurasia. Elevated dPb concentrations in the North Atlantic Ocean prior to the global-phase out of leaded gasoline (1986-2021) suggests ~5-fold higher fluxes from the North Atlantic in the late 1980s relative to 2015/2016, explaining the widespread contamination of Arctic abyssal sediments with Pb.

Lead (Pb) is a toxic element, and uptake by marine biota and incorporation into the food chain[1,2] form a pathway of exposure to humans[3,4]. The natural geochemical cycles of Pb in the ocean have been markedly perturbed by anthropogenic emissions[5,6]. Hundred-thousands of tons of Pb have been emitted into the atmosphere since the Phoenician and Roman times, particularly as a result of industrialisation, by melting of ores, combustion of coal, and the use of leaded gasoline[7,8], and resulted in large-scale deposition of Pb into the surface ocean[6]. The North Atlantic is among the regions most affected by atmospheric Pb deposition due to extensive use of Pb in North America and Europe[6,9]. With the emerging awareness of adverse health effects associated with exposure to Pb[10,11], the global phase-out of leaded gasoline between 1986 and 2021, and reductions in industrial emissions of Pb[12,13] resulted in a decrease in dissolved Pb (dPb) concentrations in surface waters of the North Atlantic Ocean[9,14]. However, current surface dPb concentrations in the Atlantic Ocean of ~20–40 pmol L$^{-1}$ (pM)[15,16] remain above pre-industrial levels (~15 pM

for surface waters in the North Atlantic Ocean before the 1850s as per ref. 14), indicating that the Pb emissions are yet to return to pre-industrial levels.

The marine biogeochemical cycle of Pb in the Arctic and Atlantic Oceans is governed by its high particle reactivity[16,17]. The tendency for Pb to absorb onto particle surfaces[18,19], particularly organic debris[20,21], results in short residence times of <1 year in particle-rich surface waters[22,23] and swift export to depth[17,24]. Residence times of dPb in intermediate and deep Atlantic waters are considerably longer (~20 years in the Eastern Arctic Ocean[25]), suggesting the possibility for long-range transport of North Atlantic Pb to the adjacent Arctic Ocean as part of the thermohaline circulation. Abyssal sediments in the Central Arctic show contamination with concentrations exceeding 30 mg kg$^{-1}$ as a result of atmospheric deposition and advection of Atlantic Water through the Fram Strait and across the Barents Sea[26,27]. This represents a ~3-fold enrichment compared to Pb concentrations of more pristine sediments, for example, near Novaya Zemlya (~10 mg kg$^{-1}$[27]). More

[1]Technical University of Braunschweig, Braunschweig, Germany. [2]GEOMAR Helmholtz Centre for Ocean Research Kiel, Kiel, Germany. [3]Department of Earth Science & Engineering, Imperial College London, London, UK. [4]Grantham Institute for Climate Change and the Environment, Imperial College London, London, UK. [5]NIOZ Royal Netherlands Institute for Sea Research, Den Burg, The Netherlands. [6]Centre for Isotope Research, University of Groningen, Groningen, The Netherlands. [7]Department of Earth, Ocean and Atmospheric Sciences, University of British Columbia, Vancouver, BC, Canada. [8]British Antarctic Survey, Cambridge, UK. [9]Present address: Flanders Marine Institute (VLIZ), Ostend, Belgium. ✉e-mail: stephan.krisch@tu-braunschweig.de

explicitly, surface sediments underlying Atlantic Water in the Eastern and Central Arctic Ocean are ~2–3-fold enriched above pre-industrial levels[28]. This indicates that atmospheric Pb deposition into the North Atlantic Ocean and subsequent advection of Atlantic Water across the Arctic-Atlantic gateways have been a prominent supply of dPb to the Arctic Ocean in the past. The first observations of dPb in surface (<15 m) waters of the Fram Strait and the Barents Sea Opening in 2012 indicated ongoing Pb transport from anthropogenic sources with Atlantic Water into the Arctic Ocean[29]. The lack of sub-surface data has so far precluded the assessment of subsurface contributions to Arctic-Atlantic dPb exchange and how water mass transport influences dPb concentrations in the Central Arctic Ocean.

In this study, we investigate the processes controlling the dPb distribution in the Arctic-Atlantic gateways based on full water column surveys in the Fram Strait (GN05, 21 July–1 September 2016), Barents Sea Opening (GN04, 6–9 October 2015) and Canadian Arctic Archipelago (GN02/GN03, 10 August–24 September 2015). We present flux calculations concerning the import and export of dPb at the Arctic-Atlantic gateways. Combined, our results show that the Arctic Ocean is a net sink for anthropogenic Pb derived from the Atlantic. Our research establishes a baseline for future investigations concerning changes in Arctic-Atlantic dPb fluxes.

## Results and discussion
### Dissolved Pb distributions
The study region was sampled over the full water column for dPb at 44 stations (Fig. 1), targeting Arctic-Atlantic exchange of water masses: 27 stations across the Fram Strait (Supplementary Fig. 1), 7 stations across the Barents Sea Opening (Supplementary Fig. 1), and 10 stations along the Parry Channel in the Canadian Arctic Archipelago (Supplementary Fig. 2). Sampling and analyses for dPb were conducted using trace element clean methods and followed GEOTRACES protocols[30] (see "Method" section for details on sampling and analyses). Water mass definitions follow Rudels et al.[31].

Dissolved Pb concentrations in surface water of the Fram Strait showed an west-to-east gradient with increasing concentrations from the Greenlandic coast (4.5 ± 2.0 pM for <50 m depth at stations 20–23, $n = 31$) towards Svalbard (17.9 ± 2.1 pM for <50 m depth at stations 1–2, $n = 6$) (Fig. 2 and Supplementary Fig. 3) indicating increasing anthropogenic impacts towards the eastern Fram Strait, in agreement with observations in 2012[29]. A subsurface maximum in the Atlantic Water of the northward-directed West Spitsbergen Current (16.7 ± 2.0 pM for depths between 50 and 176 m at stations 1–4, $n = 20$) indicates the strongest signal from historic Pb emissions through advection of sub-surface waters from the North Atlantic Ocean. A gradient in dPb concentrations was also observed in the Barents Sea Opening (Supplementary Fig. 4), yet with considerably lower dPb concentrations near Svalbard (4.0 ± 1.6 pM at station 147, $n = 11$, excl. 4 datapoints below limit of detection, and one outlier of 48 pM at bottom depth presumably derived from local sediment resuspension), suggesting a local influence from Arctic Ocean outflow with the Sørkapp Current (also termed 'East Spitsbergen Current' as per ref.[32]). Concentrations of dPb in the water column beyond the Svalbard shelf (16.0 ± 3.1 pM at station 153–173, $n = 84$) are in agreement with observations of dPb in Atlantic Water of the West Spitsbergen Current, and indicate the presence of Atlantic Water from the Norwegian Atlantic Slope Current[33] and influence from anthropogenic sources. Local recirculation of Atlantic Water in the Fram Strait[34,35], with potential temperatures of >2 °C west of 0°E[31], results in sub-surface dPb maxima between 50–400 m depth in the southbound East Greenland Current (12.0 ± 2.1 at stations 6, 14–16 and 26, $n = 27$). East Greenland Current Surface Waters (<50 m) and intermediate and deep waters are generally depleted in dPb to concentrations of ~6 pM (Fig. 2) due to advection of low dPb waters from the Central Arctic Ocean[36,37], with comparatively weak influence of atmospheric Pb deposition[38,39] reflecting the remoteness of the Arctic to major Pb emission sources in North America and Eurasia[40,41].

Waters in the Canadian Arctic are generally depleted in dPb, suggesting a limited influence from anthropogenic sources or high

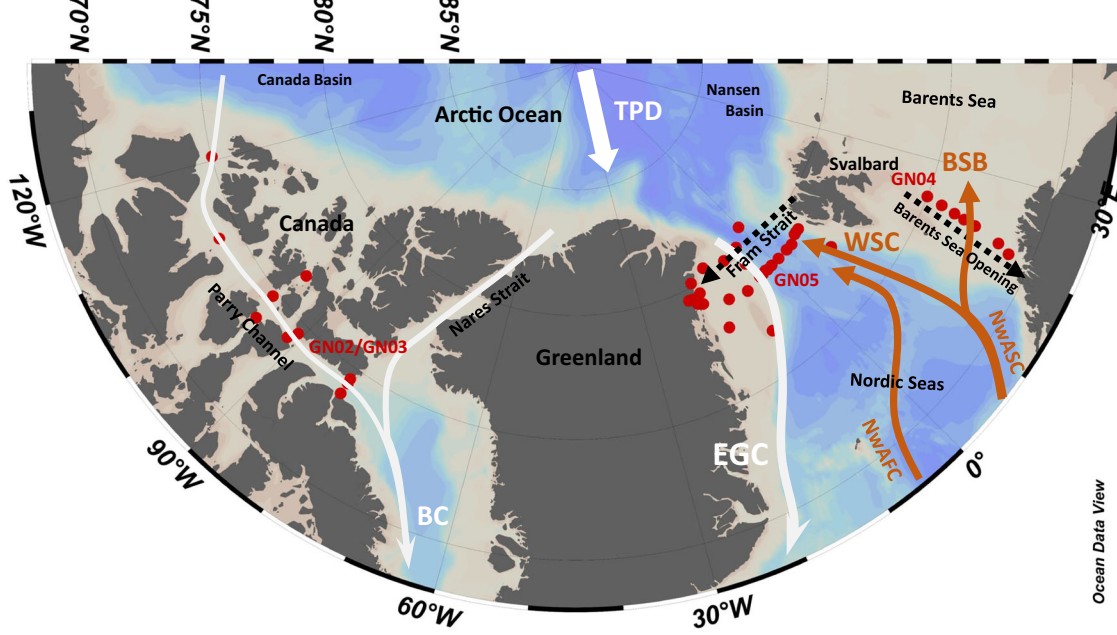

**Fig. 1 | Overview of the study region and sample locations.** Schematic of current flow paths, and location of stations (red dots) sampled during GEOTRACES expeditions GN05 (2016) in Fram Strait, GN04 (2015) in the Barents Sea Opening, and GN02/GN03 (2015) in the Canadian Arctic Archipelago. Black, dotted arrows highlight the sections which are shown in Fig. 2 (Fram Strait, GN05) and in Supplementary Fig. 4 (Barents Sea Opening, GN04). For a depiction of the CAA section, we refer the reader to ref. 43. Warm, saline (cold, fresher) Atlantic-derived (Arctic-derived) currents are depicted in bold-orange (bold-white): Norwegian Atlantic Front Current (NwAFC), Norwegian Atlantic Slope Current (NwASC), Barents Sea Branch (BSB), West Spitsbergen Current (WSC), Transpolar Drift (TPD, including Polar Surface Water), East Greenland Current (EGC), Baffin Current (BC). Station numbers are indicated in Supplementary Figs. 1 and 2. Figure produced by Ocean Data View[98].

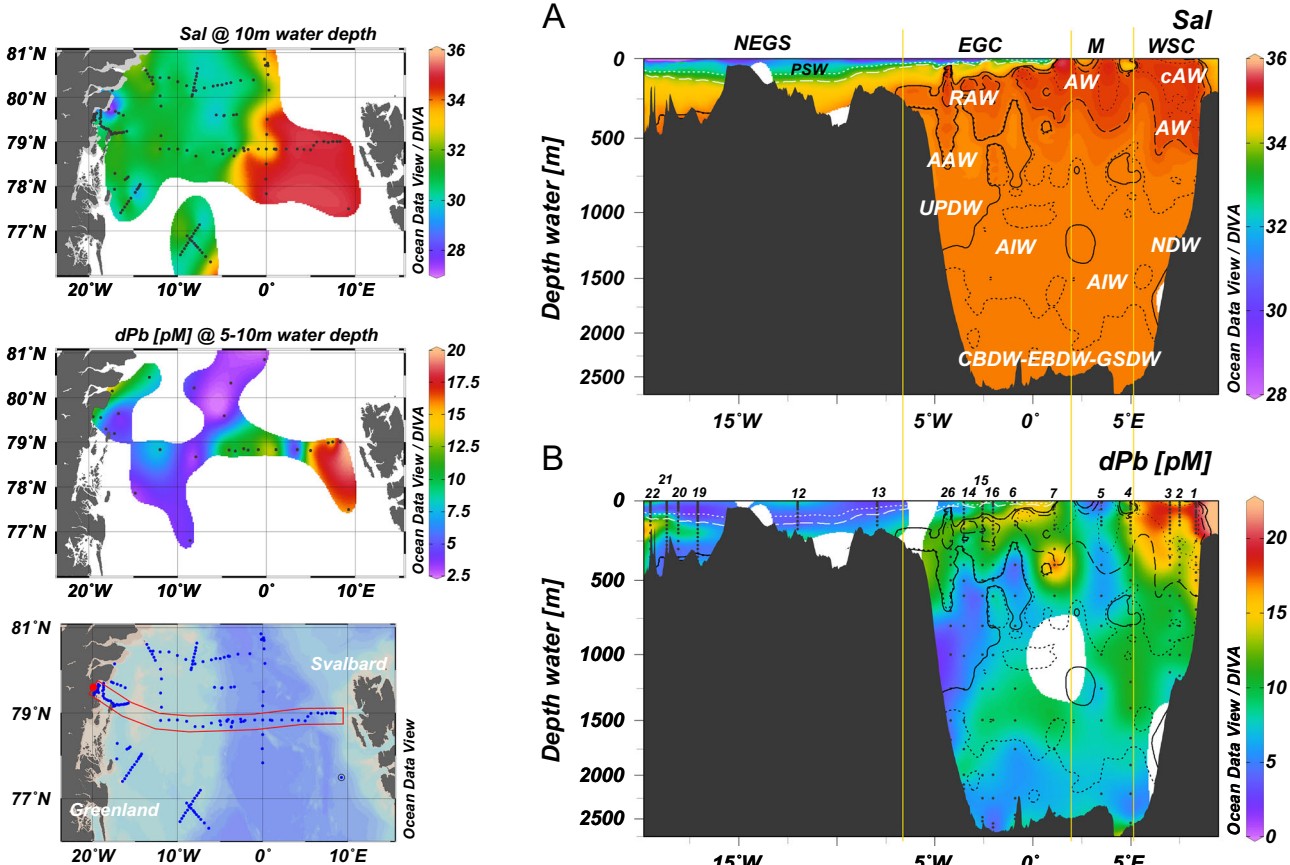

**Fig. 2 | Distributions of salinity and dissolved Pb in Fram Strait.** Section profiles of **A** salinity (Sal, top right) and **B** dissolved Pb (dPb, in pmol L$^{-1}$, pM, bottom right) across Fram Strait at 79°N from coastal Greenland ( > 15°W) to Svalbard (9°E). Surface salinity (at 10 m depth, top left), surface dPb concentrations (at 5–10 m depth, middle left), and a map of the study region highlighting stations of CTD measurements (blue dots) and the location of the cross section (in red) are show in the left column. Black dots in dPb profiles indicate sample depths. Station numbers indicated above dPb section profile. Water mass definitions by salinity, temperature, and density (CTD measurements) as per ref. 31: Atlantic Water (AW),

Recirculating Atlantic Water (RAW), Arctic Atlantic Water (AAW), Polar Surface Water (PSW), Upper Polar Deep Water (UPDW), Arctic Intermediate Water (AIW), Nordic Seas Deep Water (NDW), and Canada Basin Deep Water/Eurasian Basin Deep Water/Greenland Sea Deep Water (CBDW-EBDW-GSDW). Vertical solid lines differentiate between the West Spitsbergen Current (WSC; >5°E), Middle section ('M', 2–5°E), East Greenland Current (EGC, 2°E–6.5°W) and waters in the NE Greenland Shelf (NEGS, >6.5°W) following ref. 100. Figure generated using Ocean Data View and DIVA gridding calculations[98] and RTopo-2.0.1 bedrock topography (30-arc seconds resolution)[99].

---

scavenging intensities upstream[42,43] compared to the Fram Strait and the Barents Sea Opening. Surface waters in the upper 40 m of the Parry Channel are the strongest dPb depleted waters in the Arctic-Atlantic gateways with concentrations as low as 1.8 pM in the Viscount Melville Sound (CAA8 as per ref. 43), indicating advection of low dPb ('pristine') Canada Basin waters, including sea ice meltwater and Siberian riverine discharge[42,43]. Upper halocline waters holding contributions of Pacific origin (10.0 ± 3.7 pM between 40–90 m, $n = 6$) that entered the Arctic Ocean through the Bering Strait, and lower halocline waters (3.9 ± 1.5 pM between 120 and 300 m, $n = 7$) including contributions from Atlantic water are the main sources of dPb in the north-western parts of the Parry Channel (Stations CAA8 and CB1 as per refs. 42,43). The Barrow Strait sill (125 m depth)[44] acts as a natural barrier between the Arctic-influenced north-western parts of the Parry Channel and the Atlantic-influenced south-eastern part of the channel (stations CAA1-CAA3), and prevents the transport of dPb-enriched Atlantic waters with enhanced anthropogenic Pb concentrations from the Labrador Sea ( ~ 17–35 pM) into the Arctic Ocean[43].

Observations of dPb in Atlantic Water of the West Spitsbergen Current and the Barents Sea Opening indicate a general trend of decreasing concentrations from the North Atlantic Ocean towards the Central Arctic Ocean. Strong signals of anthropogenic Pb deposition are observed in Atlantic Water of the Irminger and Iceland Basins

(29.4 ± 6.5 pM, $n = 123$, 2010 data from GEOTRACES expedition D354)[45], decreasing across the Greenland and Norwegian Sea[29] towards the SE Arctic Ocean Nansen Basin (9.6 ± 4.8 pM at stations 32–58, $n = 30$, Supplementary Table 4). This decrease of dPb is driven by scavenging removal by elevated particle abundance from primary production in surface waters[29] as Atlantic Water is transported across the high-latitude North Atlantic and towards the Arctic Ocean. Arctic Atlantic Water, formed by cooling and freshening of Atlantic Water in the Arctic Ocean[46], is found below East Greenland Current Surface Water at depths between 175–800 m and constitutes the most dPb-depleted water mass in Fram Strait (5.3 ± 2.3 pM, $n = 35$) (Fig. 2). Scavenging processes in the Arctic Ocean inflow regions, particularly on the productive Barents Sea shelf[27,47], and boundary scavenging as Arctic Atlantic Water traverses the Arctic Ocean[25], facilitate efficient removal of dPb. The accumulated impact of scavenging is evident by the correlation between dPb concentration and mode age of Arctic Atlantic Water in the Arctic Ocean (Supplementary Fig. 5). Mode ages refer to the most probable mean age of a water mass[48]. The decreasing correlation of dPb concentrations and mode ages (0.80 $R^2$) suggests first-order removal kinetics (i.e., concentration-depending scavenging intensity) as Arctic Atlantic Water is advected across the Central Arctic. This suggests the transit time of Arctic Atlantic Water to play a crucial role in the efficiency of dPb removal from Atlantic-derived waters in

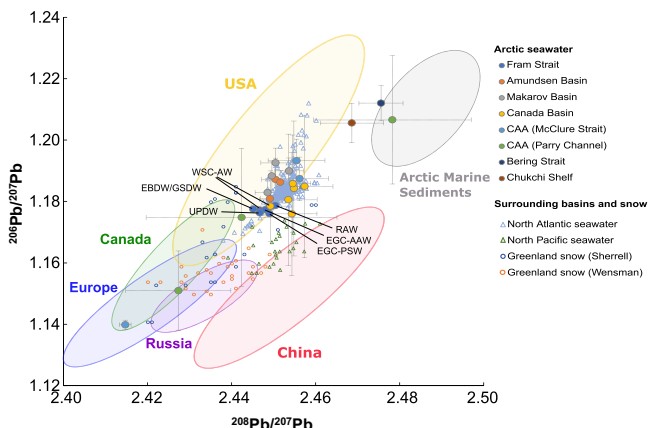

**Fig. 3 | Isotope composition of dissolved Pb in Arctic seawater and Pb isotope composition of potential source regions.** Three-isotope plot of dissolved Pb (dPb) isotope composition ($^{206}$Pb/$^{207}$Pb vs. $^{208}$Pb/$^{207}$Pb) for seawater samples collected in the Arctic Ocean (filled dots), and seawater samples from the North Atlantic Ocean (blue triangles, ref. [15]) and Pacific Ocean (green triangles, ref. [101]) and snow samples from Greenland as per Sherrell et al. (green dots, ref. [7]) and Wensman et al. (yellow dots, ref. [102]) (outlined symbols). Fram Strait samples (dark blue dots) are labelled as West Spitsbergen Current Atlantic Water (WSC-AW; one datapoint masked by Canada Basin sample), Recirculating Atlantic Water (RAW), East Greenland Current Polar Surface Water (EGC-PSW) and Arctic Atlantic Water (EGC-AAW), Upper Polar Deep Water (UPDW) and Eurasian Basin Deep Water/ Greenland Sea Deep Water (EBDW/GSDW). Arctic Ocean seawater dPb isotope composition is indicated by orange (Amundsen Basin), grey (Makarov Basin), and yellow (Canada Basin) dots. Dissolved Pb isotope composition of the Canadian Arctic Archipelago (CAA) Mc Clure Strait (light blue dots), the CAA Parry Channel (green), and the Bering Strait (black) and Chukchi Shelf (brown) are also indicated. The coloured ellipses in the background represent the Pb isotope composition of potential sources of Pb from countries neighbouring the Arctic Ocean as well as 'pristine' marine sediments collected in the Arctic Ocean basins. Canadian (green ellipse), European (blue), Russian (purple) and US (yellow) sources are inferred from aerosol Pb measurements in the respective countries between 1994–1999[49,103], and 2001–2009[104]. Chinese aerosol Pb measurements (red ellipse) were from 1994–1999[49], 2003–2005[105], and 2007–2009[106]. Arctic marine sediment Pb isotope composition (grey ellipse) is from refs. [107,108]. Error bars represent two standard deviations ($2\sigma$) from replicate measurements. Replicate measurements were not available for Fram Strait samples, and the Canadian Arctic Archipelago (CAA) McClure Strait sample, so error bars in this case represent measurement uncertainty ($2\sigma$).

the Arctic Ocean before eventually returning back into the North Atlantic Ocean.

## Sources of lead

Measurements of dPb isotope ratios, $^{206}$Pb/$^{207}$Pb and $^{208}$Pb/$^{207}$Pb, show the influence of American and Eurasian Pb emissions in the Arctic-Atlantic gateways of the Fram Strait and the Canadian Archipelago (Fig. 3). The values of $^{206}$Pb/$^{207}$Pb and $^{208}$Pb/$^{207}$Pb in pooled water samples of the West Spitsbergen Current are, despite the concentration differences, almost identical to the East Greenland Current, suggesting the dominance of one source or one mixture of sources of dPb to Arctic and Atlantic waters in the Fram Strait (Supplementary Table 6). West Spitsbergen Current Atlantic Water ($^{206}$Pb/$^{207}$Pb ratio of $1.1782 \pm 0.0007$ and $^{208}$Pb/$^{207}$Pb ratio of $2.4483 \pm 0.0015$, $n = 3$; dPb concentration of $15.4 \pm 2.3$ pM at stations S1–S4, $n = 41$), East Greenland Current Arctic Atlantic Water (1.1776 and 2.4480, $n = 1$; $5.3 \pm 2.3$ pM, $n = 35$) and Recirculating Atlantic Water (1.1789 and 2.4503, $n = 1$; $12.0 \pm 2.1$ pM, $n = 27$) fall within the range of Pb isotope compositions of aerosols from the United States. North American sources mainly emitted Pb into the atmosphere with $^{206}$Pb/$^{207}$Pb ratios varying between 1.16 and 1.22 and $^{208}$Pb/$^{207}$Pb between 2.43 and 2.46[49] owing

partly to the use of ores from the Mississippi Valley with $^{206}$Pb/$^{207}$Pb ~ 1.33 and $^{208}$Pb/$^{207}$Pb ~ 2.51[50] for the production of leaded gasoline. Westerly winds transported US aerosols and resulted in Pb enrichment of the North Atlantic Ocean[6] with US isotope signature (~ 1.16 to 1.21 for $^{206}$Pb/$^{207}$Pb and ~2.44 to 2.46 for $^{208}$Pb/$^{207}$Pb for the entire water column[15]). The similarity between the Fram Strait and North Atlantic Pb isotope signatures suggests that the main source of dPb in the Fram Strait is from historic emissions in the US and advected into the region from the North Atlantic Ocean, although likely with some local contribution from a low $^{206}$Pb/$^{207}$Pb and $^{208}$Pb/$^{207}$Pb source as indicated by the Pb isotope composition in Fram Strait being near the lower end of observations (≤10th percentile) from Atlantic Water in the North Atlantic and Arctic Oceans (Supplementary Fig. 6).

Dissolved Pb in the Canadian Arctic (Parry Channel & McClure Strait) spans a much wider range in Pb isotope compositions compared to the Fram Strait and the North Atlantic (Fig. 3), with $^{206}$Pb/$^{207}$Pb and $^{208}$Pb/$^{207}$Pb as low as $1.1402 \pm 0.0015$ and $2.4146 \pm 0.0014$ ($n = 2$). These low $^{206}$Pb/$^{207}$Pb and $^{208}$Pb/$^{207}$Pb values can be explained by a much stronger influence of Eurasian Pb emissions[42]. Indeed, Europe emitted Pb with $^{206}$Pb/$^{207}$Pb ratios between 1.10 and 1.16 and $^{208}$Pb/$^{207}$Pb between 2.37 and 2.43[49] through the consumption of leaded gasoline produced by Pb ore from Broken Hill in Australia ($^{206}$Pb/$^{207}$Pb ~ 1.04, ref. [51]). A similar $^{206}$Pb/$^{207}$Pb signature emerged from Pb emissions into the atmosphere from Russia (~1.03 to 1.15, ref. [52]) and Canada which included smelting (~0.99 to 1.07, refs. [53,54]). Canadian sources are, however, likely less important compared to Eurasian Pb supply. This is because Eurasian air masses are the dominant Pb supply to the region during winter and early spring, where aerosol Pb concentrations in the Canadian Arctic are up to 10-fold higher[38,55], and the rather localised nature of Pb emissions in the Canadian Arctic[53,56]. Interestingly, although Fram Strait values fall within the range of North Atlantic dPb composition, mean values of dissolved $^{206}$Pb/$^{207}$Pb and $^{208}$Pb/$^{207}$Pb were significantly lower compared to the North Atlantic Ocean ($p < 0.05$) and suggests some influence from Eurasian besides North American sources[57] in the Fram Strait. Dissolved $^{206}$Pb/$^{207}$Pb and $^{208}$Pb/$^{207}$Pb signatures in Fram Strait Upper Polar Deep Water ($^{206}$Pb/$^{207}$Pb of 1.1765 and $^{208}$Pb/$^{207}$Pb of 2.4466, $n = 1$; dPb concentration of $5.1 \pm 2.3$ pM, $n = 27$), and Eurasian Basin and Greenland Sea Deep Water (1.1777 and 2.4451, $n = 1$; $6.2 \pm 2.0$ pM, $n = 25$) confirm the prevalence of US sources with contributions of Eurasian-sourced dPb in intermediate and deep waters of the Fram Strait. While the overall homogeneity of the dPb isotope composition may be related to the pooling process, where variations in Pb isotope composition at discrete water depths might be 'diluted' or 'masked', evidence from the North Pacific and South Atlantic suggest reversible scavenging and vertical transport of particles may lead to migration of anthropogenic Pb signatures, characterised by relatively low $^{206}$Pb/$^{207}$Pb and $^{208}$Pb/$^{207}$Pb values, into deep waters[58,59]. This process may also play a role in the Fram Strait, given high particle fluxes in the region[60,61] and suggests anthropogenic Pb has penetrated into deep waters as observed in the Arctic Ocean Canada Basin[42], thus explaining anthropogenic Pb signatures of abyssal sediments in the Fram Strait region[26].

## Dissolved Pb flux calculations

We performed detailed flux calculations of dPb across the Arctic-Atlantic gateways of the Fram Strait, Barents Sea Opening, and Canadian Arctic to assess whether the Arctic Ocean is a sink of North Atlantic dPb between September 2015 and August 2016. The uncertainty is calculated as one standard deviation ($1\sigma$) of monthly variations in dPb fluxes following ref. [62]. For details on the calculations see Supplementary Notes. By our definition, negative values indicate an Arctic flux toward the North Atlantic Ocean ('Arctic export'), while positive values indicate a flux from the North Atlantic to the Arctic Ocean ('Arctic import').

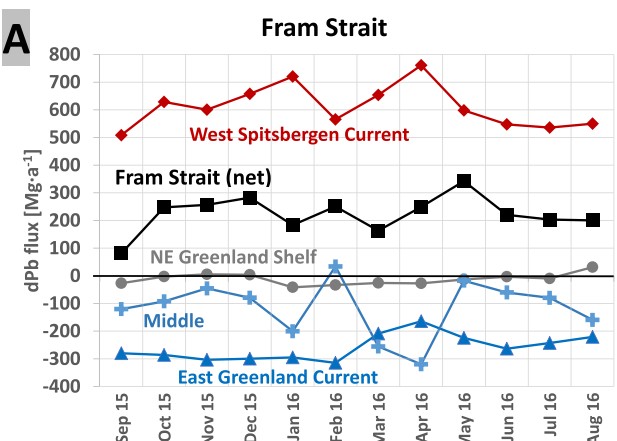

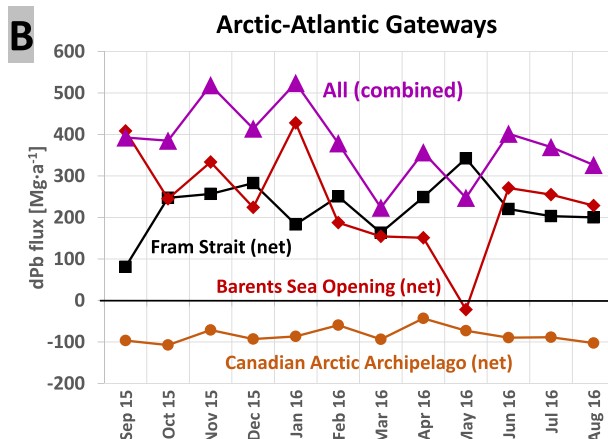

**Fig. 4 | Monthly mean estimates of dissolved Pb fluxes (in Mg a⁻¹) between September 2015 and August 2016. A** Dissolved Pb (dPb) fluxes across Fram Strait as part of the West Spitsbergen Current (red diamonds), the East Greenland Current (blue triangles), the Middle section (light blue crosses), the NE Greenland Shelf (grey dots), and Fram Strait fluxes combined (Fram Strait (net), black squares). **B** Summary of the combined dPb fluxes across the Arctic-Atlantic gateways (purple triangles) including net fluxes across Fram Strait (black squares), the Barents Sea Opening (red diamonds) and the Canadian Arctic Archipelago (brown dots). Positive values indicate an Atlantic flux into the Arctic Ocean. Negative values indicate an Arctic flux toward the North Atlantic Ocean. Visualised from Supplementary Tables 9 and 10.

Our calculations show that the Fram Strait is the most important gateway for Arctic-Atlantic exchange of dPb, owing to enhanced intermediate and deep-water volume transport across the Fram Strait (Fig. 4), which is restricted by bathymetry in the Barents Sea Opening and the Canadian Arctic. The northbound West Spitsbergen Current is the dominant source of dPb from the North Atlantic to the Arctic Ocean with a net northward flux of 611 ± 74 Mg a⁻¹ or ~60% of all Arctic import. Advection across the Barents Sea Opening further contributes 239 ± 116 Mg a⁻¹, or ~40%, to the net flux of Atlantic dPb into the Arctic Ocean. Combined, the Arctic Ocean receives a flux of 849 ± 130 Mg a⁻¹ from the North Atlantic Ocean. The southbound East Greenland Current is the major southerly current and transports −259 ± 44 Mg a⁻¹ of dPb from the Arctic Ocean towards the North Atlantic Ocean across the Fram Strait. Transport across the central parts of the Fram Strait (2–5°E, termed Middle section) adds a further ~30% (−116 ± 97 Mg a⁻¹) to the total Arctic dPb export of −387 ± 83 Mg a⁻¹ across the Fram Strait. Dissolved Pb transport on the Northeast Greenland Shelf, strongly influenced by 'near-pristine' Central Arctic Ocean surface waters[36], is only a minor contributor to Arctic dPb export across the Fram Strait (−12 ± 20 Mg a⁻¹). The Canadian Arctic is a minor gateway for Arctic dPb export and contributes a further −84 ± 18 Mg a⁻¹ to Arctic dPb export through the gateways of Nares Strait (−53 ± 8 Mg a⁻¹), Lancaster Strait (−31 ± 16 Mg a⁻¹), and Jones Sound (−0.5 ± 1.3 Mg a⁻¹) (Fig. 4 and Supplementary Table 10). Transport of Arctic dPb across the Canadian Arctic comprises only ~18% to the combined Arctic dPb export flux of ~471 ± 81 Mg a⁻¹ across the Arctic-Atlantic gateways. Combined, our results suggest that the Arctic Ocean is a net sink of 378 ± 85 Mg a⁻¹ of North Atlantic dPb in 2015/2016. This makes the North Atlantic Ocean a similarly important source as riverine discharge of dPb onto the Arctic Ocean shelves (344 ± 222 Mg a⁻¹), yet small in comparison to atmospheric Pb deposition into the surface Arctic Ocean in the early 2010s (1729 ± 1296 Mg a⁻¹) (Supplementary Information for details on the calculation).

### Seasonal variations

Seasonal variations in volume transport rates across the Arctic-Atlantic gateways may affect the flux of dPb from the North Atlantic Ocean into the Arctic Ocean on a seasonal basis. To determine seasonal changes in net Arctic import of Atlantic dPb, we investigate monthly variations in dPb fluxes across the Arctic-Atlantic gateways. Owing to the lack of seasonal dPb data in the Arctic-Atlantic gateways, we assume dPb

concentrations to remain constant in water masses over the course of the year. Monthly variations in dPb fluxes across the Fram Strait suggests a net flux of Atlantic dPb into the Arctic Ocean throughout the year (range: 81–342 Mg a⁻¹) (Fig. 4). Monthly dPb fluxes show a seasonal cycle with a sharp minimum in late summer (~81 Mg a⁻¹ for September), elevated fluxes in between October and December (~260 Mg a⁻¹) and a strong increase in Atlantic dPb transport across the Fram Strait towards May (342 Mg a⁻¹). The main driver in the seasonal variation of net Arctic dPb import across the Fram Strait is the northbound dPb transport with the West Spitsbergen Current and alterations between northward and southward transport across the central parts of the Fram Strait (between 2 and 5 °E, 'middle section'), likely a consequence of seasonal variations in local recirculation of Atlantic Water in the Fram Strait[34,35]. The advection of Atlantic dPb across the Fram Strait and the Barents Sea Opening seems to be coupled. Months of strong transport of Atlantic dPb across the Barents Sea Opening and into the Arctic Ocean coincide with decreased dPb transport across the Fram Strait, and vice versa (Supplementary Table 9). This suggests intra-annual variations also in Pb scavenging and export to depth in the Arctic-Atlantic gateways of the Fram Strait and the Barents Sea, driven by an alternating current regime, on top of seasonal change in the efficiency of primary production-induced scavenging of dPb from surface waters[29,47]. Combined, our calculations suggest a continuous net flux of Atlantic dPb into the Arctic Ocean across the Arctic-Atlantic gateways of Fram Strait, Barents Sea Opening and Canadian Arctic throughout the year (range: 224–524 Mg a⁻¹) with a maximum in winter (486 ± 51 Mg a⁻¹ for November–January) and a minimum in spring (276 ± 58 Mg a⁻¹ for March–May) and modest dPb fluxes through summer and autumn (375 ± 27 Mg a⁻¹ between June and October).

Due to the lack of seasonal dPb data in the Fram Strait and Barents Sea Opening, the exact change in monthly dPb fluxes cannot be investigated. For example, the influence of winter upwelling and sea ice meltwater on dPb concentrations in near-surface waters remains to be investigated, for the study region or anywhere else in the Arctic. Further, our dPb flux calculation across the Fram Strait and the Barents Sea Opening utilised volume flux data from 2005–2006. Quantification of the influence of changes in volume fluxes on Arctic-Atlantic dPb exchange lacks recent volume flux data in individual components for the Fram Strait and Barents Sea Opening post-2005/2006, which is currently not available. However, as >80% of dPb transport across the

Fram Strait and Barents Sea Opening occurs in subsurface waters (i.e., Atlantic, Intermediate and Deep Waters), given long-residence times of dPb in these waters of ~20 years[25,63] we expect subsurface dPb transport to be relatively independent from seasonal change in surface dPb concentrations. Concerning inter-annual change in volume fluxes, annual-mean Atlantic Water advection across the Greenland-Scotland-Ridge and towards the Arctic Ocean has remained stable between 1993 and 2016[64] and suggests that the connectivity between North Atlantic-sources and the Fram Strait and Barents Sea Opening dPb fluxes has not substantially changed.

## Atlantic dPb fluxes in the past

The Arctic Ocean has likely been a much more prominent sink for Pb in the past century, given considerably higher Pb concentrations in the North Atlantic Ocean following elevated deposition fluxes into the surface ocean after the introduction of leaded gasoline in the 1930s[6]. Time series of seawater dPb near Bermuda inferred from coral Pb/Ca content suggest surface concentrations in the western North Atlantic Ocean as high as 210 pM in response to anthropogenic Pb emissions in the 1970s[14]. High Pb concentrations were also observed in surface waters of the Eastern North Atlantic Ocean, for example in the Gulf of Biscay off the coast of France and Spain in 1990 (~180 pM at 46.2–47.3°N, 6.5–7.5°W, ref. 65). Concentrations decreased in the decades thereafter following international measures to reduce Pb emissions into the atmosphere[6,9], and in the 2010s dPb concentrations in the North Atlantic Ocean (~20–30 pM[15,45]) approached natural background levels of ~10–15 pM[14]. It is thus likely that the advection of Atlantic waters across the Arctic-Atlantic gateways resulted in considerably higher fluxes of Atlantic dPb into the Arctic Ocean between the 1970s and 2000s compared to our estimate of net 378 ± 85 Mg a$^{-1}$ for 2015/2016.

Assuming that the volume fluxes across the Arctic-Atlantic gateways were comparable between the past, and 2005–2006 and 2015–2016, the ~4.5-fold enrichment of dPb in Atlantic Water of the high latitude North Atlantic Ocean in 1989 (~110 pM at <500 m depth, 60°N/20°W) relative to 2014 (~25 pM at <500 m depth, 57°N/28°W)[15] would result in a net dPb flux of 1.8 ± 0.4 Ga a$^{-1}$ from the Atlantic into the Arctic Ocean. If we instead assume that inferred dPb concentrations of ~200 pM in the North Atlantic near Bermuda of the 1970s (Kelly et al.[14]) were representative of the high-latitude North Atlantic Ocean, then this flux estimate increases to 3.4 ± 0.7 Gg a$^{-1}$ for the 1970s. Integrated over the time span (Supplementary Fig. 7), these flux estimates suggest a total Atlantic dPb flux of ~75 Gg into the Arctic Ocean between 1970 and 2015/2016. This flux estimate likely represents an over-estimation for the Atlantic dPb flux into the Arctic Ocean, given the proximity of the Bermuda region and the high latitude North Atlantic Ocean to major Pb emissions such as from the US and Europe[6,66] and scavenging of dPb during Atlantic Water advection towards the Fram Strait and the Barents Sea Opening[29]. Yet, our estimate is helpful to understand the scale of Atlantic dPb transport into the Arctic Ocean.

## Fate of Arctic dPb import

The high Atlantic Pb flux into the Arctic Ocean is underlined by Pb contamination found in abyssal sediments underlying Arctic Atlantic Water[26,28]. Scavenging by intense primary production on the Barents Sea Shelf[47] and subduction of Atlantic Water with high particle load beneath the Arctic halocline[67] are likely key processes governing the fate of Atlantic dPb fluxes in the Arctic Ocean. Surface sediments of the Fram Strait and the adjacent Southeast Nansen Basin show the highest Pb concentrations among the Arctic Ocean sediments[26-28] with concentrations of ~35 mg kg$^{-1}$ in the Arctic Ocean inflow region northeast of Svalbard[27]. This represents a ~2–3-fold enrichment above near-pristine (~10 mg kg$^{-1}$, e.g., near Novaya Zemlya[27]) and pre-industrial values (~15 mg kg$^{-1}$[28]) and the average Pb content of Earth's upper

continental crust (~17 mg kg$^{-1}$[68]). Fram Strait and South East Nansen Basin Pb concentrations are thus close to the consensus-based threshold effect concentration of ~36 mg kg$^{-1}$[69] above which adverse health effects to benthic organisms may occur[70].

Using the Fram Strait-Southeast Nansen Basin region, we estimate the accumulated Pb content. We assume a Pb enrichment of abyssal sediments between 2 mg kg$^{-1}$[28] and 10 mg kg$^{-1}$[26], an average benthic surface mixed layer depth of between 2 and 5 cm[26,28], and an average marine sediment density of 1.70 g cm$^{-3}$[71]. Scaled to the surface area of the region (~10% of the Arctic Ocean basin, 0.5 Mio km$^2$[72]), our calculations suggest that the Fram Strait-Southeast Nansen Basin sector of the Arctic Ocean has accumulated between 34 and 425 Gg of Pb from anthropogenic emissions. This Pb sedimentation flux is comparable to the anthropogenic Pb input into rivers, lakes, and oceans in the early 1980s (97–277 Gg a$^{-1}$)[73], but only a fraction of the tens of thousands of gigagrams of anthropogenic Pb emissions into the atmosphere[40,74]. Yet, the suggested Pb sedimentation flux in the Fram Strait-Southeast Nansen Basin region may considerably exceed the Arctic-Atlantic dPb flux estimate of ~75 Gg between 1970 and 2015/2016. Elevated dPb concentrations >5-fold above the natural background in the North Atlantic Ocean since at least 1900[14] suggest that there also has been a net flux of Atlantic dPb into the Arctic Ocean prior to the 1970s, possibility in the same range as ~75 Gg post-1970. Additionally, other sources such as atmospheric Pb deposition[75,76], advection of sea ice enriched in atmospheric and lithogenic Pb[77,78] and meltwater release near the Arctic-Atlantic gateways[79] likely also contributed to sediment Pb enrichment in the Fram Strait-Southeast Nansen Basin region.

With dPb concentrations in the North Atlantic Ocean declining[14,15], we expect the Atlantic dPb flux into the Arctic Ocean to decline in the future as well. Lead thus serves as a prime example for the effectiveness of environmental regulations to reduce anthropogenic perturbations in the Arctic-Atlantic marine environment. However, climate change-driven alterations to the current regime in the region[80,81], may in future affect dPb transport across the Arctic-Atlantic gateways. For example, an increase in Atlantic Water volume transport across the Fram Strait, as projected in response to sea ice decline[80], may increasingly counteract anticipated declines in Atlantic dPb transport into the Arctic Ocean. This would lead to a larger fraction of Atlantic dPb that is scavenged and transferred to depth in the Arctic Ocean or incorporated into the Arctic food chain rather than being scavenged and transferred to sediments or taken up by biology in the Atlantic Ocean. Even if the volume flux of Atlantic Water into the Arctic Ocean does not increase, the observed shoaling of the Atlantic Water layer in the Central Arctic Ocean[82] means that a higher fraction of the Atlantic dPb may be taken up by primary producers further north compared to scavenging from subsurface Atlantic Water through interaction with the continental slope in Fram Strait and the Southeast Nansen Basin[25,27].

While abyssal sediments are the ultimate sink of anthropogenic Pb emissions into the ocean, the fate of anthropogenic Pb deposited in Arctic shelf sediments is less clear. Productive shelf regions such as the Barents Sea play a crucial role in the fate of Atlantic dPb in the Arctic Ocean, and are more relevant than abyssal sediments in terms of their bioaccumulation potential due to the stronger benthic-pelagic coupling. In shelf regions, Pb is largely delivered to the seafloor through sinking biogenic material, which is remineralised and potentially released back into the water column[83]. In this respect, increased re-working of Arctic shelf sediments[84] from increasing storm activity driving water column mixing on the shelf and upwelling of near-shelf waters under climate change[85], may support the mobilisation of the sedimentary Pb pool[16]. The result would likely be an increase in the dPb fluxes across the Arctic Ocean shelf breaks and towards the Central Arctic Ocean, with unknown consequences for Arctic Pb levels in seawater and marine biota.

## Methods

### Sampling

A detailed description of the sampling process during PS100/GN05 and PS94/GN04 was published in refs. 86,87, respectively. In brief, the Fram Strait and the Barents Sea Opening were sampled on-board the research vessel Polarstern during GEOTRACES expeditions GN05 (PS100, 21 July–6 September 2016) and GN04 (PS94, 17 August–15 October 2015), respectively. Both expeditions followed GEOTRACES standards for sampling and sample handling[30].

Sampling in the Fram Strait utilised a powder-coated aluminium GEOTRACES frame (Seabird, equipped with a SBE 911 CTD) equipped with 24 × 12 L GoFlo bottles. Sampling in the Barents Sea Opening was conducted with the NIOZ titanium frame (equipped with a Seabird SBE 911) and 24 × 24 L ultra-trace metal clean polypropylene samplers. Both expeditions sampled the water column to full depth. Vertical, full-depth profiles of salinity, temperature and pressure were conducted at high resolution on both expeditions. Sub-sampling was conducted in an over-pressured class 100 ultraclean laboratory container immediately after CTD recovery. Samples were filtered into pre-cleaned LDPE (low-density polyethylene) bottles using Acropak™ 500/1000 capsules (0.8/0.2 μm pore size; Pall Corp.) on GN05 and Sartobran® 300 capsules (< 0.2 μm pore size; Sartorius) on GN04. Samples were acidified to pH 1.9 (GN05) or pH 1.8 (GN04) using ultrapure hydrochloric acid from ROMIL (GN05) or Seastar Chemicals Inc. (GN04).

Details on GN02/GN03 sampling in the Canadian Arctic Archipelago on board the CCGS Amundsen (10 July-1 October 2015) have been published with refs. 42,43.

### Dissolved Pb analyses

Dissolved Pb samples from expedition GN05 were analysed via high-resolution inductively coupled plasma-mass spectrometry (HR-ICP-MS) after solid-phase extraction exactly as per Rapp et al.[88]. In brief, 10 mL of sample aliquots were UV-digested and preconcentrated using an automated SeaFAST system (SC4 DX SeaFAST pico; ESI). All reagents were prepared in deionized water (>18.2 MΩ cm; Milli-Q, Millipore) from high-purity chemicals. Sub-boiled nitric acid (SpA grade, ROMIL) was used to prepare a 1 M nitric acid for sample elution. Ammonium acetate buffer was prepared from glacial acetic acid and ammonium hydroxide (Optima grade, Fisher Scientific). The 10-fold preconcentrated samples were then analysed by HR-ICP-MS (Thermo Fisher Element XR) via external calibration (standard addition) with Pb standard (Inorganic Ventures Inc.). Dissolved Pb samples from expedition GN04 were analysed via HR-ICP-MS (Thermo Finnigan Element 2) after solid-phase extraction following Gerringa et al.[89]. Briefly, 30 mL of sample aliquots were UV-digested and preconcentrated using an automated SeaFAST system (SeaFAST pico; ESI). The preconcentration was conducted with ammonium hydroxide and sub-boiled glacial acetic acid solution (Suprapur grade, Merck) prepared in deionized water (>18.2 MΩ cm; Milli-Q, Millipore). The 40-fold preconcentrated samples were eluted in 1.5 M Teflon-distilled sub-boiled nitric acid (Suprapur grade, Merck). Calibration was via standard addition (external calibration) from a Pb standard (TraceCERT, Sigma-Aldrich).

The analysis of GN02/GN03 samples of the Canadian Arctic Archipelago has been described in detail by ref. 43.

### Dissolved Pb isotope composition analysis

Samples for the analysis of dissolved Pb isotope composition from GEOTRACES expedition GN05 were pooled according to water mass definitions of Atlantic Water, Arctic Atlantic Water and Recirculation Atlantic Water in the Fram Strait as per Rudels et al.[31] (Supplementary Table 6). Samples were analysed in the ISO class 6 clean rooms and the mass spectrometry facilities of the MAGIC Laboratories at Imperial College London via Multi-Collector Inductively Coupled Plasma Mass Spectrometry (MC-ICP-MS) following the methodology exactly of Griffiths et al.[90]. Briefly, 0.7–2.3 L of seawater, depending on its Pb

concentration, were used to pre-concentrate and separate Pb for isotopic analysis using the Nobias Chelate PA-1 resin and anion exchange chromatography. The Pb eluted in the final step of chromatography was split in two aliquots: 1/3 of it was spiked with a $^{204}$Pb–$^{207}$Pb double spike, and the remaining 2/3 was left unspiked. Both aliquots were doped with a solution of NIST Standard Reference Material (SRM) 997 Tl to obtain an elemental ratio of Pb:Tl of 1:3 to enable correction of the measured Pb isotope ratios for instrumental mass bias using the double spike technique[91]. The analysis of GN02/GN03 dissolved Pb isotope samples of the Canadian Arctic Archipelago has been described in detail by ref. 42.

### Reporting summary

Further information on research design is available in the Nature Portfolio Reporting Summary linked to this article.

## Data availability

All data used throughout this publication are accessible online. Physical oceanography data can be obtained from: https://doi.pangaea.de/10.1594/PANGAEA.871025 (PS100/GN05 large CTD[93]), https://doi.pangaea.de/10.1594/PANGAEA.871030 (PS100/GN05 clean CTD[94]), https://doi.pangaea.de/10.1594/PANGAEA.859558 (PS94/GN04 large and clean CTD[95]). Canadian Arctic Archipelago volume fluxes are calculated from a simulation with the ANHA12 configuration of the NEMO ocean model, which can be obtained from: https://canadian-nemo-ocean-modelling-forum-commuity-of-practice.readthedocs.io/en/latest/Institutions/UofA/Configurations/ANHA12/index.html. Pb data can be obtained from: https://doi.org/10.1594/PANGAEA.933431 (dissolved Pb from GN05[96]), https://doi.org/10.1594/PANGAEA.968782 (Pb isotope composition from GN05[97]), https://doi.org/10.25850/nioz/7b.b.jc (dissolved Pb from GN04[89]). Source data for Figs. 2 and 3 can be found in Source Data. The map of the study region (Fig. 1), and the section profiles and surface distributions (Fig. 2) were made by SK with Ocean Data View software and DIVA gridding calculations[98] and RTopo-2.0.1 bedrock topography (30-arc seconds resolution)[99]. The three-isotope plot of dissolved Pb (dPb) isotopic composition (Fig. 3) was made by AO using Microsoft Excel and Inkscape. The plots of monthly dPb fluxes (Fig. 4) were made by SK with Microsoft Excel (version 2019, https://office.microsoft.com/excel). Source data are provided with this paper.

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

## Acknowledgements
The authors thank the captain and crew of the RV Polarstern, operated by the Alfred Wegener Institute, Helmholtz Centre for Polar and Marine Research[92]. We are grateful for the help received from Micha Rijkenberg (NIOZ) for organisation and sampling, and to Patrick Laan (NIOZ) for trace element analyses of GN04. Thanks also to Sven Ober (NIOZ) for the operation and maintenance of the PRISTINE sampling system, and to Ari-dane Gonzalez (IUEM) and Michael Staubwasser (University of Cologne) for their assistance in trace element clean sampling during GN04. Many thanks to Pablo Lodeiro, Florian Evers, Nicola Herzberg, and Jaw Chuen Yong (all GEOMAR) for organisation and assistance during sampling of GN05. Special thanks to Tim Steffens and Stéphane Roig (both GEO-MAR) for help in trace metal analysis of GN05 samples. We are grateful to Takamasa Tsubouchi (Japan Meteorological Agency) for sharing the volume transport data. German cruise work and analyses of Fram Strait samples were financed by GEOMAR and the German Research Foundation (DFG award number AC 217/1-1 to E.A.). The writing and publication of this manuscript was supported by the Technical University of Braunschweig. The Dutch participation was funded by the Netherlands Organization for Scientific Research (NWO contract number 822.01.018 to L.G.). Arianna Olivelli was supported by the NERC Natural Environment Research Council (NE/S007415/1).

## Author contributions
S.K. conceived the study. Sampling was conducted by S.K. and L.G. Pb analyses were conducted by S.K., R.M., L.G., and A.O. Statistical analysis was conducted by S.K. and B.R. The first draft of the manuscript was written by S.K. with all authors (S.K., E.A., A.O., R.M., L.G., B.R.) contributing to its final version.

## Funding

## Competing interests
The authors declare no competing interests.
