## [Peer Review File · Nature Communications]

The Arctic Ocean is a net sink for anthropogenic lead deposited into the Atlantic Ocean

Corresponding Author: Dr Stephan Krisch

Version 0:

Reviewer comments:

Reviewer #1

(Remarks to the Author)

Krisch and colleagues present data from a cruise to the Fram Strait region in which they measured dissolved Pb concentrations and isotope ratios. These, along with previously measured seawater fluxes, are used to develop flux estimates of dPb between the Atlantic and Arctic basins. The data show that there is a net positive flux of Pb from the Atlantic into the Arctic, where the Pb is likely scavenged and deposited on the seafloor. Considering historic emissions of Pb from leaded gasoline in the United States, the flux of dPb from the Atlantic into the Arctic was likely much higher in the past.

The data appear to be of high quality, and uncertainties are quantified. The interpretations are well-supported by data and the paper is clear and easy to read. There are a few minor English usage errors throughout the manuscript, which will likely be corrected during copy-editing. Regarding the substance of the paper, I only have some minor comments, mainly related to the figures.

Fig. 2

Depth indication in middle-left panel has a typo. Please make the lower-left panel the same size/scale as the other left-hand panels, since they appear to cover the same geographic extent.

Figure 3. Please clarify whether each source ellipse indicates bedrock/sediment ("natural") sources of lead and/or coal/ore/smelting/incineration/fuels ("pollution") sources of lead. For instance, the "China" ellipse only extends partway into the range of Chinese coal data for Pb, and does not contain the Chinese desert dust sources at all. I really think the ellipses need to accurately reflect the source regions and be more explicit about what is included. Data citations for the Greenland snow and also the potential source areas would also be helpful here.

Figure 4 would benefit from better labeling (e.g. "Fram Strait" over left panel and "Arctic-Atlantic Gateways" over right panel, and explicit legends for the data lines in each panel rather than making the reader derive all the information from looking back and forth between the caption and the figure. Also, I note that "FS (net)" is labeled in both panels but has different colors. If these are the same lines, please use the same colors/symbols for clarity.

This is a minor comment, but I think the figures would look more cohesive if a similar color scheme were used throughout. Since Ocean Data View provides a rainbow palette, perhaps Fig. 3 could use more vibrant (less pastel) colors and Fig. 4 could use shades that more closely align with those in Fig. 2.

Line 181: Probably better to use "US" here to avoid confusion between North American (inclusive of Canada and Mexico) and USA sources.

Line 296 and elsewhere: Please consider using the English spelling of Novaya Zemlya

Line 305: consider also providing this value in Gg for consistency with previous units, for ease of comparison.

Line 324: This deserves additional context. Surely the literature can give some estimates of the Pb levels considered harmful to marine life. Even if future Pb release from shelf sediments is difficult to quantify, some sense of what Pb concentrations are known to be damaging to ecosystems would be helpful here or earlier in the text, for context.

Supplementary table 3: the values for 206/207 and 208/207 should be calculated and added to the table for Consensus value A.

Supplementary figures 3 and 4: Please make the lower-left panel the same size as the others; there is no reason for it to be so small with all this white space surrounding it. In addition, these multi-panel figures would be improved by adding letter labels (A,B,C,D) for easy reference.

Reviewer #2

(Remarks to the Author)

Krisch et al. presented lead concentrations and isotopic compositions across three key oceanic gateways connecting the Atlantic Ocean and the Arctic Ocean. They reported significantly elevated lead concentrations in the Fram Strait and Barents Sea Opening, and much lower concentrations in the Greenland Shelf and Canadian Arctic Archipelago gateways. Following the direction of water mass transport in these regions, the authors argue that the Fram Strait and Barents Sea Opening act as primary conduits for anthropogenic lead entering the Arctic from the Atlantic. Within the Arctic, they propose that water masses gradually lose lead through scavenging processes as they age, which explains the lower lead concentrations observed in water masses exiting the Arctic and returning to the Atlantic (While I generally agree with this interpretation, I would add that the remoteness of the Arctic from major emission sources likely also contributes to the lower lead levels in Arctic-derived water masses).

The authors went on to estimate the flux of lead transported from the Atlantic to the Arctic, showing that it is significant relative to riverine inputs to the Arctic Ocean. They also extrapolated this flux to the 1970s, when leaded gasoline emissions were at their peak. Additionally, they compared the total flux of lead entering the Arctic via oceanic transport to the accumulated anthropogenic lead stored in Arctic sediments.

Overall, I found the manuscript to be well-written and supported by high-quality data. As the global lead emission is rising again due to electrification, understanding the fate and pathways of lead transport across and within the environment, as well as the key sinks will be of rising importance. The pathway and sink described by Krisch et al is interesting and relevant to broad.

However, I have concerns about the final part of the manuscript, particularly the comparison between the accumulated lead in sediments and the flux of Atlantic-transported lead. While the authors have made commendable efforts to trace the fate of lead after entering the Arctic Ocean, the available data can only confidently demonstrate that the sedimentary lead inventory is significantly larger than the Atlantic-derived flux. This comparison implies that the sediment lead may include a portion of the transported lead, but may not be dominant. As there is a much larger portion of lead flux over toning the signal in the sediment, it is difficult to determine with confidence the fate of the Atlantic transported lead. Moreover, the high sedimentary lead content appears to be the only independent line of evidence supporting their calculated flux, aside from physical oceanographic data, which is not fully independent as they used them into the calculation. Clarifying the 'sedimentary lead' line of evidence is critically important in supporting their overall argument. I therefore recommend that the authors either clarifying the sediment inventory and flux comparison, or think a bit deeper and more explicitly define the limits of the conclusions that can be drawn from their sediment–flux comparison.

A second concern relates to the discussion on climate implications, which I have elaborated on in the specific comments below.

Detailed comments below:

Line 39: I agree that not everywhere in North Atlantic returns to natural, but better to be more specific on your geography as some study detected natural pb in north Atlantic.

Supp Fig 3: bottom left panel better note where is Greenland and where is Svalbard, in order to make the readers geographically oriented

Line 86-87: What is similar? Have a similar subsurface maximum at similar depth?

Line 142-143: Fig 3 is oriented differently with description – description is by current but figure is by basin. Without aligned geographical orientation, I cannot figure out where are the samples representing West Spitsbergen Current and East Greenland Current. Can you specify what colors are each?

Line 164: these numbers does not look variable at all with such a low SD

Line 211: I don't think 'pristine' is the correct word, maybe 'near pristine'?

Line 284-287: I think it is not appropriate to assume the high latitude water have the same concentration as inferred Bermuda due to the fact it is farther away from major emitters at the point in time. Therefore, I recommend to add a disclaimer like 'which is possibly an overestimation but is helpful to show the scale of the transport'.

Line 301-305: I am confused here.

The study calculates that the flux inferred from the Abyssal sediments, which is 34-425 Gg in total (assume an integration over the past half century where gasoline was at its peak). The advective transport from Atlantic Ocean was 3.4Gg/a, as estimated in line 286, even 20 years of peak gasoline emission (which is an overestimation as not every year emits as much as 1970s) could only accumulate 64Gg, in the low end of the estimated accumulation. Does it mean that the atmosphere input remains dominant Pb flux to Arctic? If that is the case, the high lead concentration in the sediments should be a primary result of increased gasoline lead emissions in 1960s-1980s rather than the Atlantic-Arctic transport?

Last paragraph on implications under climate change:

I read the change in current direction and water mass layering a bit come out from blue. I do not have data but expect wherever the lead is, regardless of in the Atlantic or Arctic, will partly join the food chain. More Pb in Atlantic versus Arctic

does not seem to directly impact the overall Pb – possibly only affecting who gets more. And I also think it unfair to say the seafood in one ocean more important than another. Therefore, I really think the argument needs more thoughts. For the sediment benthic flux part, I agree with your points. Just a small suggestion: shelf sediments might be more relevant than abyssal sediments in terms of bioaccumulation, as they are located closer to productive coastal and shelf regions and are more accessible to the marine life that humans depend on. It might be helpful to draw stronger connections to the implications for the food chain and ecosystem services in these areas.

Version 1:

Reviewer comments:

Reviewer #2

(Remarks to the Author)

It is the second time I am reviewing Krisch et al. 's manuscript regarding the transport of Atlantic lead to the Arctic Ocean. Before commenting, I must make a disclaimer that as I could not find the 'manuscript with changes' file in the reviewer's file pack. As a result, I could not trace where the edits are located, but to believe the authors have made respective adjustments to those cosmetic comments. The authors have made good improvements in response to my earlier major comments, particularly regarding the flux budget calculation and the implications in the context of climate change. The revised manuscript incorporates budget calculations showing the total flux of lead transported from Atlantic to Arctic while acknowledging other sources. This is highly appreciated, as it illustrates the scale of the advective transport flux in the total pool. Another major improvement is that the authors have refined their discussion of implications by considering both Atlantic and Arctic, as well as both shelf seas and the deep ocean, thereby providing a more balanced perspective. As I mentioned in the previous reviewer report, the global lead emission is rising again due to electrification, understanding the fate and pathways of lead transport across and within the environment, as well as the key sinks will be of rising importance. I therefore support the publication of this manuscript in Nature Communications.

Mengli

Reviewer 1

General Comments:

Krisch and colleagues present data from a cruise to the Fram Strait region in which they measured dissolved Pb concentrations and isotope ratios. These, along with previously measured seawater fluxes, are used to develop flux estimates of dPb between the Atlantic and Arctic basins. The data show that there is a net positive flux of Pb from the Atlantic into the Arctic, where the Pb is likely scavenged and deposited on the seafloor. Considering historic emissions of Pb from leaded gasoline in the United States, the flux of dPb from the Atlantic into the Arctic was likely much higher in the past.

The data appear to be of high quality, and uncertainties are quantified. The interpretations are well-supported by data and the paper is clear and easy to read. There are a few minor English usage errors throughout the manuscript, which will likely be corrected during copy-editing. Regarding the substance of the paper, I only have some minor comments, mainly related to the figures.

Thank you very much for your time and the positive feedback you provide. Please find below our responses to your in-line comments.

Specific Line Comments:

Fig. 2.: Depth indication in middle-left panel has a typo. Please make the lower-left panel the same size/scale as the other left-hand panels, since they appear to cover the same geographic extent.

Done.

Figure 3. Please clarify whether each source ellipse indicates bedrock/sediment (“natural”) sources of lead and/or coal/ore/smelting/incineration/fuels (“pollution”) sources of lead. For instance, the “China” ellipse only extends partway into the range of Chinese coal data for Pb, and does not contain the Chinese desert dust sources at all. I really think the ellipses need to accurately reflect the source regions and be more explicit about what is included. Data citations for the Greenland snow and also the potential source areas would also be helpful here.

We have added the requested information regarding the Pb sources areas. The ellipses refer to Pb isotope composition of aerosols as summarized by (Bollhöfer and Rosman, 2000, 2001; Lee *et al.*, 2007) from the years 1994-1999 and 2003-2005. We have now also included references to the Pb isotope composition of North Atlantic and North Pacific seawater and Greenlandic snow in the caption.

Figure 4 would benefit from better labeling (e.g. “Fram Strait” over left panel and “Arctic-Atlantic Gateways” over right panel, and explicit legends for the data lines in each panel rather than making the reader derive all the information from looking back and forth between the caption and the figure. Also, I note that “FS (net)” is labeled in both panels but has different colors. If these are the same lines, please use the same colors/symbols for clarity.

Done.

This is a minor comment, but I think the figures would look more cohesive if a similar color scheme were used throughout. Since Ocean Data View provides a rainbow palette, perhaps Fig. 3 could use more vibrant (less pastel) colors and Fig. 4 could use shades that more closely align with those in Fig. 2.

We have thought about aligning all figures to the same colour scheme. However, since several different programmes were used for generating the figures of the main manuscript and in the supplement, aligning the colours to one scheme is not directly possible. However, we will contact the Nature Communications graphic editor and raise your suggestion upon acceptance of this manuscript.

Line 181: Probably better to use “US” here to avoid confusion between North American (inclusive of Canada and Mexico) and USA sources.

Done.

Line 296 and elsewhere: Please consider using the English spelling of Novaya Zemlya.

Done.

Line 305: consider also providing this value in Gg for consistency with previous units, for ease of comparison.

Done.

Line 324: This deserves additional context. Surely the literature can give some estimates of the Pb levels considered harmful to marine life. Even if future Pb release from shelf sediments is difficult to quantify, some sense of what Pb concentrations are known to be damaging to ecosystems would be helpful here or earlier in the text, for context.

We thank the reviewer for making us aware of this shortcoming. We have revised the corresponding section to include a statement on Pb threshold concentrations (lines 302-307):

“...This represents a ~2-3-fold enrichment above near-pristine (~10 mg·kg⁻¹ e.g. near Novaya Zemlya (Kohler, Kull, et al., 2022)) and pre-industrial values (~15 mg·kg⁻¹ (Gobeil et al., 2001)) and the average Pb content of Earth’s upper continental crust (~17 mg·kg⁻¹ (Rudnick and Gao, 2003)). Fram Strait and South East Nansen Basin Pb concentrations are thus close to the consensus-based threshold effect concentration of ~36 mg·kg⁻¹ (MacDonald, Ingersoll and Berger, 2000) above which adverse health effects to benthic organisms may occur (Long et al., 1995).”

Please note that while there are threshold Pb concentrations in literature that are indicative of adverse health effects to e.g. benthic organisms (e.g. (Long et al., 1995; MacDonald, Ingersoll and Berger, 2000)), it is difficult to use threshold concentrations precisely as indicator of toxicity. Toxicity depends on the presence of pollutant species and their bio-availability, and ideally refers to a specific organism (e.g. the most prevailing or most Pb-sensitive in the study region) (Chapman et al., 1999, 2003). This is because of species-specific uptake mechanisms, seasonality in uptake, acclimation and adaptation, and synergistic or antagonistic effects with other elements that may increase or decrease the toxicity of Pb (e.g. (Ansari, Marr and Tariq,

2003; Chapman *et al.*, 2003). Also, changes in sediment chemistry such as from alterations in sediment oxygen concentrations – may they be seasonal, interannual or decadal – may increase or decrease the toxicity of metal pollutants through remobilisation (e.g. (O’Connor and Paul, 2000)). For the study region, there is no information available regarding benthic species-composition and sediment chemistry including the presence of Pb pollutant species, their bioavailability or seasonality. We have hence ‘only’ stated the consensus-based threshold effect concentrations of (MacDonald, Ingersoll and Berger, 2000) which is a sediment quality guideline indicating adverse health effects from exposure to Pb, and still frequently used today e.g. (Cortis *et al.*, 2024; Jezycki *et al.*, 2024; Bonso *et al.*, 2025)).

Supplementary Table 3: the values for 206/207 and 208/207 should be calculated and added to the table for Consensus value A.

Thank you for this comment. We have considered calculating the ‘missing’ Pb isotope ratios for the masses 206/207 and 208/207, also for the Supplementary Table S6. While we see the benefit of adding these, the tables S3 and S6 represent a data compilation of measured Pb isotope ratios from literature. Owing to the source of data, we hence did not calculate corresponding masses of Pb that were not measured as we think this would not be appropriate particularly with respect to reference material, and may be misleading to the reader.

Supplementary figures 3 and 4: Please make the lower-left panel the same size as the others; there is no reason for it to be so small with all this white space surrounding it. In addition, these multi-panel figures would be improved by adding letter labels (A,B,C,D) for easy reference.

Done.

Reviewer 2

General Comments:

Krisch *et al.* presented lead concentrations and isotopic compositions across three key oceanic gateways connecting the Atlantic Ocean and the Arctic Ocean. They reported significantly elevated lead concentrations in the Fram Strait and Barents Sea Opening, and much lower concentrations in the Greenland Shelf and Canadian Arctic Archipelago gateways. Following the direction of water mass transport in these regions, the authors argue that the Fram Strait and Barents Sea Opening act as primary conduits for anthropogenic lead entering the Arctic from the Atlantic. Within the Arctic, they propose that water masses gradually lose lead through scavenging processes as they age, which explains the lower lead concentrations observed in water masses exiting the Arctic and returning to the Atlantic (While I generally agree with this interpretation, I would add that the remoteness of the Arctic from major emission sources likely also contributes to the lower lead levels in Arctic-derived water masses).

The authors went on to estimate the flux of lead transported from the Atlantic to the Arctic, showing that it is significant relative to riverine inputs to the Arctic Ocean. They also extrapolated this flux to the 1970s, when leaded gasoline emissions were at their peak. Additionally, they compared the total flux of lead entering the Arctic via oceanic transport to the accumulated anthropogenic lead stored in Arctic sediments.

Overall, I found the manuscript to be well-written and supported by high-quality data. As the global lead emission is rising again due to electrification, understanding the fate and pathways of lead transport across and within the environment, as well as the key sinks will be of rising importance. The pathway and sink described by Krisch et al is interesting and relevant to broad. However, I have concerns about the final part of the manuscript, particularly the comparison between the accumulated lead in sediments and the flux of Atlantic-transported lead. While the authors have made commendable efforts to trace the fate of lead after entering the Arctic Ocean, the available data can only confidently demonstrate that the sedimentary lead inventory is significantly larger than the Atlantic-derived flux. This comparison implies that the sediment lead may include a portion of the transported lead, but may not be dominant. As there is a much larger portion of lead flux over toning the signal in the sediment, it is difficult to determine with confidence the fate of the Atlantic transported lead. Moreover, the high sedimentary lead content appears to be the only independent line of evidence supporting their calculated flux, aside from physical oceanographic data, which is not fully independent as they used them into the calculation. Clarifying the 'sedimentary lead' line of evidence is critically important in supporting their overall argument. I therefore recommend that the authors either clarifying the sediment inventory and flux comparison, or think a bit deeper and more explicitly define the limits of the conclusions that can be drawn from their sediment–flux comparison. A second concern relates to the discussion on climate implications, which I have elaborated on in the specific comments below.

The reviewer is thanked for the comments which we addressed more specifically with the in-line comments below. We included a note that the remoteness of the Arctic from major Pb emissions likely contribute to lower levels of Pb in Arctic-derived water masses (lines 100-101). We revised the section regarding the calculated Pb inventory in sediments of the Fram Strait-Nansen Basin region and agree with the reviewer that other sources besides Atlantic dPb advection may be present. However, we have a different viewpoint than the reviewer with respect to our claim that Atlantic dPb transport into the Arctic Ocean and accumulation in basin sediments is not supported by literature. The following publications are relevant:

(Gobeil *et al.*, 2001) were the first to report elevated Pb concentrations in sediments underlying Atlantic Water in the Arctic Ocean. The authors observed a Pb enrichment in the SE Nansen Basin close to the continental shelf slope which is extrapolated to the Eurasian Basin to derive a total anthropogenic Pb flux between 9 and 48 megatons. Scaled to the Fram Strait-SE Nansen Basin region would have a sediment inventory between 3 and 16 megatons (3000-16000 Gg). This value is in the range of anthropogenic Pb emissions into the atmosphere of ca. 20000 Gg before 1980 (Nriagu, 1979), thus likely a considerable overestimation. We have not commented on (Gobeil *et al.*, 2001) estimate as we deem their calculations to be heavily biased by the assessment of only three stations in the Nansen and Amundsen Basin (two stations in the Nansen Basin) and one station in the Greenland Sea (south of the Arctic Ocean), however their estimate does highlight accumulation of anthropogenic Pb in Arctic basin sediments.

(Kohler, Kull, *et al.*, 2022) observed Pb deposition and accumulation in abyssal sediments of the SE Nansen Basin, and suggest scavenging in the Barents Sea and subduction below the Arctic Ocean halocline to play a crucial role in the fate of Pb advection (in line with water column Hg and Pb observations presented in (Kohler, Heimbürger-Boavida, *et al.*, 2022)).

(Smith, Moran and Macdonald, 2003) investigated on ocean-shelf interactions of ²¹⁰Pb in the Arctic Ocean and derived a residence time for Atlantic Pb of 15-20 years because of boundary scavenging as Atlantic Water is advected along the continental slope.

(Carignan, Hillaire-Marcel and De Vernal, 2008) traced Pb and Pb isotopes in Fram Strait sediments to material transport from the North Atlantic (eastern parts of Fram Strait) and from the Laptev Sea by the Transpolar Drift (western parts of Fram Strait). Where relevant, we have cited these publications more consistently.

We have also added further back-up for our claims on the fate of Pb in the Arctic Ocean, and elaborated on climate change impacts on Pb fluxes in the Arctic-Atlantic gateways. Please see our response to the in-line comments.

Specific Line Comments:

Line 39: I agree that not everywhere in North Atlantic returns to natural, but better to be more specific on your geography as some study detected natural pb in north Atlantic.

We are not aware of a study that has shown a return to natural Pb concentrations in the North Atlantic Ocean. The reviewer may refer to the publication of (Bridgestock *et al.*, 2016), where a return to natural Pb concentrations is observed in the North Atlantic Ocean, yet not achieved (e.g., the lowest fraction of anthropogenic dPb was around 40-50% of the total Pb at 20-30°N). Also (Kelly *et al.*, 2009), using Pb/Ca ratios of corals as proxy, observed a return to more natural Pb concentrations, yet this was not achieved up to the year 2000 at the end of their timeline. All other studies in the North Atlantic Ocean we are aware of such as from (Rusiecka *et al.*, 2018; Zurbrick *et al.*, 2018; Schlosser and Garbe-Schönberg, 2019; Achterberg *et al.*, 2021) suggest ongoing and strong perturbation of the natural Pb cycle through anthropogenic Pb.

Supp Fig 3: bottom left panel better note where is Greenland and where is Svalbard, in order to make the readers geographically oriented.

Done.

Line 86-87: What is similar? Have a similar subsurface maximum at similar depth?

We have clarified the corresponding line.

Line 142-143: Fig 3 is oriented differently with description – description is by current but figure is by basin. Without aligned geographical orientation, I cannot figure out where are the samples representing West Spitsbergen Current and East Greenland Current. Can you specify what colours are each?

Thank you for making us aware of this shortcoming. We have indicated the corresponding Fram Strait samples by current in our revised Figure 3.

Line 164: these numbers does not look variable at all with such a low SD.

We have now clarified that the variability did not refer to individual samples but to the range of observed Pb isotope composition, as visible in Figure 3.

Line 211: I don't think 'pristine' is the correct word, maybe 'near pristine'?

Done.

Line 284-287: I think it is not appropriate to assume the high latitude water have the same concentration as inferred Bermuda due to the fact it is farther away from major emitters at the point in time. Therefore, I recommend to add a disclaimer like ‘which is possibly an overestimation but is helpful to show the scale of the transport’.

Thank you for this comment. We have now revised the corresponding lines and included a statement on the caveats of our dPb flux estimation (now lines 289-294):

“This flux estimate likely represents an over-estimation for the Atlantic dPb flux into the Arctic Ocean given the proximity of the Bermuda region and the high latitude North Atlantic Ocean to major Pb emissions such as from the US and Europe (Pacyna and Pacyna, 2000; Boyle et al., 2014) and scavenging of dPb during Atlantic Water advection towards Fram Strait and the Barents Sea Opening (Schlosser and Garbe-Schönberg, 2019). Yet, our estimate is helpful to understand the scale of Atlantic dPb transport into the Arctic Ocean.”

Line 301-305: I am confused here. The study calculates that the flux inferred from the Abyssal sediments, which is 34-425 Gg in total (assume an integration over the past half century where gasoline was at its peak). The advective transport from Atlantic Ocean was 3.4Gg/a, as estimated in line 286, even 20 years of peak gasoline emission (which is an overestimation as not every year emits as much as 1970s) could only accumulate 64Gg, in the low end of the estimated accumulation. Does it mean that the atmosphere input remains dominant Pb flux to Arctic? If that is the case, the high lead concentration in the sediments should be a primary result of increased gasoline lead emissions in 1960s-1980s rather than the Atlantic-Arctic transport?

Thank you for making us aware of this. Based on the calculated fluxes from 1970, 1989 and 2015/2016 we have now included a rough estimate on the total (integrated) Atlantic dPb transport into the Arctic Ocean (ca. 75 Gg between 1970 and 2015/2016 (Supplementary Figure S7)). While this is the range of the suggested Pb sedimentation in the Fram Strait-SE Nansen Basin region (34-425 Gg), our 1970-2015/2016 estimate is indeed at the lower end of this range. While there has likely also been a net flux of Atlantic dPb into the Arctic Ocean prior to the 1970s (as indicative from elevated dPb concentration in Atlantic Water near Bermuda, (Kelly et al., 2009)) which may be in a similar magnitude to our 1970-2015/2016 flux, other sources such as from atmospheric deposition and sea ice meltwater release likely also contributed to Pb deposition in the Fram Strait-SE Nansen Basin region. We have thus revised the paragraph as follows (lines 317-325):

“Yet, the suggested Pb sedimentation flux in the Fram Strait-Southeast Nansen Basin region may considerably exceed the Arctic-Atlantic dPb flux estimate of ~75 Gg between 1970 and 2015/2016. Elevated dPb concentrations >5-fold above the natural background in the North Atlantic Ocean since at least 1900 (Kelly et al., 2009) suggest that there also has been a net flux of Atlantic dPb into the Arctic Ocean prior to the 1970s, possibility in the same range as ~75 Gg post-1970. Additionally, other sources such as from atmospheric Pb deposition (Kadko et al., 2016; Grotti et al., 2024), advection of sea ice enriched in atmospheric and lithogenic Pb (Hölemann, Schirmacher and Prange, 1999; Marsay et al., 2018) and meltwater release near the Arctic-Atlantic gateways (Maccali et al., 2012) likely also contributed to sediment Pb enrichment in the Fram Strait-Southeast Nansen Basin region.”

Last paragraph on implications under climate change: I read the change in current direction and water mass layering a bit come out from blue. I do not have data but expect wherever the lead is, regardless of in the Atlantic or Arctic, will partly join the food chain. More Pb in Atlantic versus Arctic does not seem to directly impact the overall Pb – possibly only affecting who gets more. And I also think it unfair to say the seafood in one ocean more important than another. Therefore, I really think the argument needs more thoughts.

For the sediment benthic flux part, I agree with your points. Just a small suggestion: shelf sediments might be more relevant than abyssal sediments in terms of bioaccumulation, as they are located closer to productive coastal and shelf regions and are more accessible to the marine life that humans depend on. It might be helpful to draw stronger connections to the implications for the food chain and ecosystem services in these areas.

The reviewer is correct. Changes to the current regime of the region such as from increases in Atlantic Water transport across Fram Strait mainly effects the distribution of dPb, i.e. a larger fraction of Atlantic dPb is then potentially scavenged and exported to depth in the Arctic compared to the Atlantic Ocean. We have modified the corresponding section in the revised version of the manuscript (lines 333-340):

“This would lead to a larger fraction of Atlantic dPb that is scavenged and transferred to depth in the Arctic Ocean or incorporated into the Arctic food chain rather than being scavenged and transferred to sediments, or taken up by biology in the Atlantic Ocean. Even if the volume flux of Atlantic Water into the Arctic Ocean does not increase, the observed shoaling of the Atlantic Water layer in the Central Arctic Ocean (Polyakov et al., 2025) means that a higher fraction of the Atlantic dPb may be taken up by primary producers further north compared to scavenging from subsurface Atlantic Water through interaction with the continental slope in Fram Strait and the Southeast Nansen Basin (Smith, Moran and Macdonald, 2003; Kohler, Kull, et al., 2022).”

We have also included a statement that productive areas such as the Barents Sea shelf play a crucial role in the fate of Pb in the Arctic with a higher bioaccumulation potential compared to abyssal sediments (lines 342-345):

“Productive shelf regions such as the Barents Sea play a crucial role in the fate of Atlantic dPb in the Arctic Ocean, and are more relevant than abyssal sediments in terms of their bioaccumulation potential due to the stronger benthic-pelagic coupling.”

Literature cited:

- Achterberg, E.P. *et al.* (2021) 'Trace Element Biogeochemistry in the High-Latitude North Atlantic Ocean: Seasonal Variations and Volcanic Inputs', *Global Biogeochemical Cycles*, 35(3), p. e2020GB006674. Available at: <https://doi.org/10.1029/2020GB006674>.
- Ansari, T.M., Marr, I.L. and Tariq, N. (2003) 'Heavy Metals in Marine Pollution Perspective—A Mini Review', *Journal of Applied Sciences*, 4(1), pp. 1–20. Available at: <https://doi.org/10.3923/jas.2004.1.20>.
- Bollhöfer, A. and Rosman, K.J.R. (2000) 'Isotopic source signatures for atmospheric lead: The Southern Hemisphere', *Geochimica et Cosmochimica Acta*, 64(19), pp. 3251–3262. Available at: [https://doi.org/10.1016/S0016-7037\(00\)00436-1](https://doi.org/10.1016/S0016-7037(00)00436-1).
- Bollhöfer, A. and Rosman, K.J.R. (2001) 'Isotopic source signatures for atmospheric lead: The Northern Hemisphere', *Geochimica et Cosmochimica Acta*, 65(11), pp. 1727–1740. Available at: [https://doi.org/10.1016/S0016-7037\(00\)00630-X](https://doi.org/10.1016/S0016-7037(00)00630-X).
- Bonso, M. *et al.* (2025) 'Accumulation of metals in resident blue mussels (*Mytilus edulis*) and exposed Asian clams (*Corbicula fluminea*) along the Scheldt estuary: A spatial and temporal investigation', *Marine Pollution Bulletin*, 213, p. 117619. Available at: <https://doi.org/10.1016/j.marpolbul.2025.117619>.
- Boyle, E.A. *et al.* (2014) 'Anthropogenic lead emissions in the ocean: The evolving global experiment', *Oceanography*, 27(1), pp. 69–75. Available at: <https://doi.org/10.5670/oceanog.2014.10>.
- Bridgestock, L. *et al.* (2016) 'Return of naturally sourced Pb to Atlantic surface waters', *Nature Communications*, 7:12921. Available at: <https://doi.org/10.1038/ncomms12921>.
- Carignan, J., Hillaire-Marcel, C. and De Vernal, A. (2008) 'Arctic vs. North Atlantic water mass exchanges in Fram Strait from Pb isotopes in sediments', *Canadian Journal of Earth Sciences*, 45(11), pp. 1253–1263. Available at: <https://doi.org/10.1139/E08-050>.
- Chapman, P.M. *et al.* (1999) 'Appropriate applications of sediment quality values for metals and metalloids', *Environmental Science and Technology*, 33(22), pp. 3937–3941. Available at: <https://doi.org/10.1021/es990083n>.
- Chapman, P.M. *et al.* (2003) 'Conducting ecological risk assessments of inorganic metals and metalloids: Current status', *Human and Ecological Risk Assessment*, 9(4), pp. 641–697. Available at: <https://doi.org/10.1080/713610004>.
- Cortis, R. *et al.* (2024) 'Ecological risk from potentially toxic element legacy contamination in sediment from the Forth and Clyde Canal, Scotland, UK', *Environmental Monitoring and Assessment*, 196(9), p. 833. Available at: <https://doi.org/10.1007/s10661-024-12995-3>.
- Gobeil, C. *et al.* (2001) 'Atlantic Water Flow Pathways Revealed by Lead Contamination in Arctic Basin Sediments', *Science*, 293(5533), pp. 1301–1304. Available at: <https://doi.org/10.1126/science.1062167>.
- Grotti, M. *et al.* (2024) 'New insights into the sources of atmospheric lead reaching the Arctic by isotopic analysis of PM10 atmospheric particles and resuspended soils', *Atmospheric Environment*, 330, p. 120541. Available at: <https://doi.org/10.1016/j.atmosenv.2024.120541>.
- Hölemann, J.A., Schirmacher, M. and Prange, A. (1999) 'Dissolved and Particulate Major and Trace Elements in Newly Formed Ice from the Laptev Sea (Transdrift III, October 1995)', *Land-Ocean Systems in the Siberian Arctic*, (October 1995), pp. 101–111. Available at: https://doi.org/10.1007/978-3-642-60134-7_11.
- Jezycki, K.E. *et al.* (2024) 'Metal accumulation in salt marsh soils along the East Coast of the United States', *Science of the Total Environment*, 922, p. 171025. Available at: <https://doi.org/10.1016/j.scitotenv.2024.171025>.
- Kadko, D. *et al.* (2016) 'Determining the pathways, fate, and flux of atmospherically derived trace

- elements in the arctic ocean/ice system', *Marine Chemistry*, 182, pp. 38–50. Available at: <https://doi.org/10.1016/j.marchem.2016.04.006>.
- Kelly, A.E. *et al.* (2009) 'Lead concentrations and isotopes in corals and water near Bermuda, 1780-2000', *Earth and Planetary Science Letters*, 283(1–4), pp. 93–100. Available at: <https://doi.org/10.1016/j.epsl.2009.03.045>.
- Kohler, S.G., Heimbürger-Boavida, L.-E., *et al.* (2022) 'Arctic Ocean's wintertime mercury concentrations limited by seasonal loss on the shelf', *Nature Geoscience*, 15, pp. 621–626. Available at: <https://doi.org/10.1038/s41561-022-00986-3>.
- Kohler, S.G., Kull, L.M., *et al.* (2022) 'Distribution pattern of mercury in northern Barents Sea and Eurasian Basin surface sediment', *Marine Pollution Bulletin*, 185:114272. Available at: <https://doi.org/10.1016/j.marpolbul.2022.114272>.
- Lee, C.S.L. *et al.* (2007) 'Heavy metals and Pb isotopic composition of aerosols in urban and suburban areas of Hong Kong and Guangzhou, South China-Evidence of the long-range transport of air contaminants', *Atmospheric Environment*, 41(2), pp. 432–447. Available at: <https://doi.org/10.1016/j.atmosenv.2006.07.035>.
- Long, E.R. *et al.* (1995) 'Incidence of adverse biological effects within ranges of chemical concentrations in marine and estuarine sediments', *Environmental Management*, 19(1), pp. 81–97. Available at: <https://doi.org/10.1007/BF02472006>.
- Maccali, J. *et al.* (2012) 'Pb isotopes and geochemical monitoring of Arctic sedimentary supplies and water mass export through Fram Strait since the Last Glacial Maximum', *Paleoceanography*, 27(1), p. PA1201. Available at: <https://doi.org/10.1029/2011PA002152>.
- MacDonald, D.D., Ingersoll, C.G. and Berger, T.A. (2000) 'Development and evaluation of consensus-based sediment quality guidelines for freshwater ecosystems', *Archives of Environmental Contamination and Toxicology*, 39(1), pp. 20–31. Available at: <https://doi.org/10.1007/s002440010075>.
- Marsay, C.M. *et al.* (2018) 'Dissolved and particulate trace elements in late summer Arctic melt ponds', *Marine Chemistry*, 204, pp. 70–85. Available at: <https://doi.org/10.1016/j.marchem.2018.06.002>.
- Nriagu, J.O. (1979) 'Global inventory of natural and anthropogenic emissions of trace metals to the atmosphere', *Nature*, 279(5712), pp. 409–411. Available at: <https://doi.org/10.1038/279409a0>.
- O'Connor, T.P. and Paul, J.F. (2000) 'Misfit between sediment toxicity and chemistry', *Marine Pollution Bulletin*, 40(1), pp. 59–64. Available at: [https://doi.org/10.1016/S0025-326X\(99\)00153-8](https://doi.org/10.1016/S0025-326X(99)00153-8).
- Pacyna, J.M. and Pacyna, E.G. (2000) *Atmospheric emissions of anthropogenic lead in Europe: Improvements, updates, historical data and projections*, *Berichte der GKSS*. Geesthacht. Available at: <https://www.osti.gov/etdeweb/servlets/purl/20148180>.
- Polyakov, I. V. *et al.* (2025) 'Atlantification advances into the Amerasian Basin of the Arctic Ocean', *Science Advances*, 11(8), p. eadq7580. Available at: <https://doi.org/10.1126/sciadv.adq7580>.
- Rudnick, R.L. and Gao, S. (2003) 'Composition of the Continental Crust', in H.D. Holland and K.K. Turekian (eds) *Treatise on Geochemistry*. Elsevier, pp. 1–64. Available at: <https://doi.org/10.1016/B0-08-043751-6/03016-4>.
- Rusiecka, D. *et al.* (2018) 'Anthropogenic Signatures of Lead in the Northeast Atlantic', *Geophysical Research Letters*, 45(6), pp. 2734–2743. Available at: <https://doi.org/10.1002/2017GL076825>.
- Schlosser, C. and Garbe-Schönberg, D. (2019) 'Mechanisms of Pb supply and removal in two remote (sub-)polar ocean regions', *Marine Pollution Bulletin*, 149, p. 110659. Available at: <https://doi.org/10.1016/j.marpolbul.2019.110659>.
- Smith, J.N., Moran, S.B. and Macdonald, R.W. (2003) 'Shelf-basin interactions in the Arctic Ocean based on ²¹⁰Pb and Ra isotope tracer distributions', *Deep-Sea Research Part I: Oceanographic*

Research Papers, 50(3), pp. 397–416. Available at: [https://doi.org/10.1016/S0967-0637\(02\)00166-8](https://doi.org/10.1016/S0967-0637(02)00166-8).

Zurbrick, C.M. *et al.* (2018) ‘Dissolved Pb and Pb isotopes in the North Atlantic from the GEOVIDE transect (GEOTRACES GA01) and their decadal evolution’, *Biogeosciences*, 15, pp. 4995–5014. Available at: <https://doi.org/10.5194/bg-2018-29>.